# Cysteine Cathepsins as Therapeutic Targets in Immune Regulation and Immune Disorders

**DOI:** 10.3390/biomedicines11020476

**Published:** 2023-02-07

**Authors:** Emanuela Senjor, Janko Kos, Milica Perišić Nanut

**Affiliations:** 1Department of Biotechnology, Jožef Stefan Institute, 1000 Ljubljana, Slovenia; 2Faculty of Pharmacy, University of Ljubljana, 1000 Ljubljana, Slovenia

**Keywords:** cysteine cathepsins, inflammation, autoimmune diseases, cancer

## Abstract

Cysteine cathepsins, as the most abundant proteases found in the lysosomes, play a vital role in several processes—such as protein degradation, changes in cell signaling, cell morphology, migration and proliferation, and energy metabolism. In addition to their lysosomal function, they are also secreted and may remain functional in the extracellular space. Upregulation of cathepsin expression is associated with several pathological conditions including cancer, neurodegeneration, and immune-system dysregulation. In this review, we present an overview of cysteine-cathepsin involvement and possible targeting options for mitigation of aberrant function in immune disorders such as inflammation, autoimmune diseases, and immune response in cancer.

## 1. Introduction

### 1.1. Cathepsins in Immune Regulation

Cysteine cathepsins (Cat) can be found not only in the lysosomes, but also in the cytoplasm, cell nucleus and extracellular space, where they are often associated with pathological conditions [1,2]. Eleven cysteine cathepsins are encoded in the human genome (Cats B, C, F, H, K, L, O, S, V, W, and X) [3]. Most of these cathepsins exhibit endopeptidase activity, with the exception of CatB and CatX, which are carboxyexopeptidases [4,5] and CatC and H, the aminopeptidases [6]. Furthermore, depending on the pH of the environment, CatB and CatH can have both endo- and exo-peptidase activity [7,8]. Cathepsins’ function is regulated at several levels—gene expression, post-translational modifications such as glycosylation, localization, and finally by endogenous inhibitors—stefins, cystatins, kininogens, and serpins [9]. Structural elements in exopeptidases enable electrostatic bonds with the C- or N-termini of the substrates and restrict their direct access to the active site [10,11]. Cathepsins are also synthesized as zymogens and are autoactivated in the acidic and reducing environment of lysosomes, by interaction with glycosaminoglycans, or by proteolytic processing with other peptidases [12]. Further processing also takes place for CatB, H, and L, which can be cleaved to disulfide-linked heavy and light chains that retain the proteolytic activity [13]. Cysteine cathepsins have been shown to be involved in the development of certain diseases such as cancer, neurodegeneration, and autoimmune diseases, where the dysregulation of cysteine-cathepsins expression or activity has been shown to be a contributing factor [14]. By understanding the mechanisms of cathepsins signaling, researchers hope to develop new therapeutic strategies, which are reviewed here.

### 1.2. Intracellular Cathepsin Signaling

Cathepsins are involved in many processes including intracellular protein turnover and degradation [15], activation of Toll-like receptors, antigen processing [16,17], and autophagy, as well as processing of various hormones and growth factors [18,19] and activation of cytotoxic effectors in granules in natural-killer (NK) cells and cytotoxic T lymphocytes (CTL) [20]. Additionally, they can trigger apoptosis by degradation of Bid and other mechanisms [21]. In physiological conditions cathepsins are confined to the endo/lysosomal environment, whereas under pathological conditions, their localization is expanded to the nucleus where they process transcription factors, facilitating cell proliferation and differentiation [22]. In the cytoplasm, they facilitate inflammation signaling and cell death. Moreover, under pathological conditions cysteine cathepsins can also be frequently found in the extracellular space [6,18,23].

### 1.3. Extracellular Cathepsin Signaling

Immune cells are the most common source of extracellular cathepsins, strengthening their connection to inflammatory disorders [24]. As mentioned above, cysteine cathepsins are active in acidic pH [25]. Inflammatory conditions in cancer and autoimmune disorders are characterized by acidification of the extracellular space, which enables the activity of cathepsins [26]. In the extracellular space, cysteine cathepsins participate in extracellular matrix remodeling [27] and processing of cytokines and chemokines [28,29], leading to disruption of signalling pathways, changes to the microenvironment, and generation of neoantigens, which in turn leads to development of multiple pathologies [24].

## 2. Targeting Cysteine Cathepsins in Different Immune Pathologies

### 2.1. Inflammation and Autoimmune Diseases

In this section, we will review the role of cysteine cathepsins in various inflammatory conditions (Figure 1, Table 1). In addition, we will present several cysteine-cathepsin inhibitors that were used in clinical trials. 

CatS cleaves the invariant chain p10 (invariant fragment, CD76, Lip10) that enables assembly of MHC class II-Ag peptide complexes and consequently plays a vital role in the regulation of MHC class II surface antigen (Ag) presentation on antigen-presenting cells (APCs) to T and B cells. CatS inhibition with selective inhibitor RO5459072 was able to block Lip10 degradation in healthy donor- and systemic lupus erythematosus (SLE) patient-derived B cells and supress the induction of proinflammatory macrophages ex vivo. [30]. Therefore, CatS was suggested to specifically improve SLE symptoms by inhibiting autoantigen presentation (Figure 1). In a mouse model of SLE, inhibition of CatS led to a decrease in infiltration and activation of splenic DC as well as the activation of CD4+ T helper and CD4-CD8- T cells [31]. In the same model CatS inhibition also reduced autoantibody production and renal deposition of immune complexes that significantly improved SLE related nephritis [31]. In a mouse model of rheumatoid arthritis (RA), pharmacological inhibition of CatS led to a decrease in disease progression and disease scoring, suggesting a potential positive effect in targeting CatS in RA [32]. Significantly higher activity of CatS was detected in tears of patients with Sjögren’s syndrome in comparison to healthy individuals and patients with autoimmune diseases [33], and also in tears of a NOD (non-obese diabetic) mouse model of Sjögren’s syndrome [34]. Therefore, CatS was proposed as a biomarker for both primary and secondary Sjögren’s syndrome [33,34]. Inhibition of CatS in patients with primary Sjögren syndrome using selective inhibitor RO5459072 resulted in decreased T-cell response toward autoantigens (SS-A, soluble substance A and SS-B, soluble substance B) and suppressed cytokine secretion by CD14+ monocytes [35]. Additionally, CatS inhibition improved salivary flux and diminished glandular inflammation in a NOD (non-obese diabetic) mouse model of Sjögren’s syndrome [33]. In the same mouse model, inhibition of CatS with a peptide-based inhibitor significantly lowered CatS activity in tears and the lacrimal gland and reduced lymphocytic infiltration in the lacrimal gland and improved tear secretion [36]. Pharmacological inhibition of CatS with selective inhibitor RO5459072 was considered as a promising approach for alleviating ocular symptoms of Sjögren’s syndrome; however, its evaluation did not pass phase II clinical trial [37]. In patients with psoriasis the expression of CatS is significantly upregulated [38] together with IL-36γ which was shown to be its major target in psoriatic tissue [28]. Several structurally different CatS inhibitors were shown to be effective in either a psoriatic or an atopic dermatitis mouse model but only one CatS (VBY-891) inhibitor successfully passed a phase I safety study for psoriasis, and the study discontinued after the phase II efficacy phase [33]. CatS is also implicated in pathology of bronchial asthma, since it was shown to be upregulated in preclinical allergic mouse models [39]. The study using CatS knockout mice highlighted its role in the development of allergic inflammation, since the inhibition of CatS led to a significant decrease in the inflammation and airway infiltration of proinflammatory immune cells [39]. However, despite significant success on animal models, none of the CatS inhibitors has entered phase III clinical trials, possibly due to species-specific differences in CatS as well as differences in the human versus mouse immune response.

CatC is involved in the regulation of immune processes and inflammation through mediation of neutrophil serine protease (NSP) maturation. Polymorphonuclear neutrophils form the first line of defence against the inbreak of microorganisms as they are the first cells recruited to the site of inflammation and infection. Their dominant effector mechanism is the secretion of NSPs, including elastase (NE), CatG, proteinase 3 (PR3), and neutrophil serine protease 4 (NSP4) [40,41]. In immature neutrophils NSPs are present in zymogen form that requires removal of dipeptide from the N-terminal by CatC [2,42]. In phagolysosomes of mature neutrophil NSPs unite with reactive oxygen species (ROS) to enable pathogen degradation [40]. Overt and uncontrolled activity of NSPs is a hallmark of a wide range of serious inflammatory conditions and their role in sepsis, acute pancreatitis, RA, chronic obstructive pulmonary disease (COPD), bronchiectasis, and cystic fibrosis is reviewed in [43] in more detail. Furthermore, the secretion of NSPs was showed to be one of the factors contributing to complications related to severe COVID-19 cases [44]. As a key regulator of NSP maturation, CatC is implicated in diseases that have excessive neutrophil infiltration as one of the causal factors for progression. One such example is GPA (granulomatosis with polyangitis) [45] where the inhibition of CatC activity led to reduced neutrophil infiltration and autoantibody priming as a consequence of decreased neutrophil chemotaxis [46] implicating CatC as a promising target for treatment of neutrophil-related inflammatory conditions [43]. Furthermore, CatC is also responsible for generation of proinflammatory chemokines [47] in activated microglia [48] that promote microglia M1 polarization and aggravate neuroinflammation via the Ca2+-dependent PKC/p38MAPK/NF-kB pathway [49]. The increase in expression of its endogenous inhibitor cystatin F (CysF), in a mouse model of multiple sclerosis (MS), was shown to reduce the activity of CatC mediating the remyelination in a chronic demyelinating disease model. In this model, the balance between CatC and CysF expression was suggested to control the demyelination and remyelination process [50]. Recent studies have shown that CatC is involved in neuroinflammation through stimulation of chemokine (CCL2 and CXCL2) production from injured neurons and glia, indicating that pharmacological inhibition of CatC activity could be one of the strategies for decreasing neuronal damage during inflammation or upon traumatic brain injury [51]. In the past two decades several reversible and irreversible inhibitors of CatC have been developed but only a handful have reached clinical testing. Some of the problems that limited further advance of other inhibitors in clinical studies were the high-level inhibition of enzyme activity necessary to achieve therapeutically significant effects [52,53], the compensatory role of CatH, and the on-target effect such as the major lack of processing of multiple-granule serine peptidases. The first irreversible CatC inhibitor that reached clinical trials was GSK-2793660. Although it was observed that GSK2793660 inhibited CatC activity but not the activity of downstream NSPs, repeated dosing led to adverse effects and the clinical trial was discontinued. Brensocatib (INS1007) is an oral reversible inhibitor of CatC that showed significant reduction of NSP activity in patients with bronchiectasis [54]. Thus far, it is the only CatC inhibitor that successfully finished clinical trials and was subsequently approved by FDA for the treatment of adult non-cystic fibrosis bronchiectasis (NCFBE) [43].

CatL has also been studied in connection to inflammatory pathologies. Substantially increased secretion of CatL from microglia has been observed 1 h after treatment with LPS, which is earlier than the upregulation of pro-inflammatory cytokines, indicating that the earlier release of lysosomal CatL in microglia may contribute to inflammatory responses [55]. Inhibition of CatL with NapSul-Ile-Trp-CHO, alleviated the microglia-mediated neuroinflammatory responses through caspase-8 and NF-κB pathways. One of the key factors in the development of neurodegenerative pathologies, such as Parkinson’s disease was shown to be microglia-mediated neuroinflammation [56], and CatL inhibition could provide more clues on how to fight excessive neuroinflammation [57]. The cells that first respond to molecules signalling neuronal damage in the central nervous system such as damage-associated molecular patterns (DAMPs), chemokines, and ATP are resident microglia. They respond by upregulating the expression and secretion of several proinflammatory cytokines and chemokines causing infiltration of astrocytes and peripheral immune cells [58]. 

Another cathepsin with a role in neuroinflammation is CatB. Through inflammasome-independent processing of procaspase-3, microglial CatB stimulates production and secretion of IL-1β [59,60]. The role of CatB in age-associated neuroinflammation is complex. On one hand its leakage from lysosomes has been shown to lead to degradation of mitochondrial transcription factor A and destabilization of mitochondrial DNA. On the other hand, CatB contributes to degradation of amyloid beta (Aβ) thus helping to maintain neuronal homeostasis [61,62]. As reviewed in [63] blood–brain-barrier-permeable, highly selective CatB inhibitors can be beneficial in regulating and maintaining neuronal homeostasis and functions during neuroinflammation and brain aging. Animal studies showed that both E64d (a cell-permeable cathepsin inhibitor) and CA-074Me (the highly specific CatB inhibitor) significantly improved Alzheimer’s disease memory loss and decreased Aβ burden. [64,65]. CatB was significantly upregulated in muscle tissues of patients with polymyositis, and in a guinea-pig model of coxsackievirus B1(CVB1)-induced polymyositis, the expression of CatB in muscles was significantly upregulated and inhibition of CatB activity with CA-074Me lowered the inflammation score and reduced infiltration of macrophages CD68(+) and TNF-α (+) cells [66]. In the same model of CVB1-induced polymyositis, inhibition of CatB activity decreased the apoptosis in muscle tissue through inhibition of Bax expression. Increased CatB expression was also detected in muscle tissue of polymyositis patients [66].

The secretion of CatX has also been associated with inflammation processes in the brain. In response to neuronal damage and inflammatory stimulus, CatX is excessively secreted from microglia and astrocytes both in culture and in vivo [67,68,69,70]. In vitro, in response to an inflammatory stimulus such as LPS, microglia cells are activated and markedly increase secretion of CatX [68,69]. When CatX was inhibited in activated microglia in vitro, using the highly selective inhibitor AMS36, a significant decrease in the production of reactive oxygen species and the pro-inflammatory cytokines was detected. Increased expression of CatX in activated microglia cells as well as in glial cells and reactive astrocytes in response to LPS was also detected in vivo. Injection of LPS unilaterally into a rat brain led to an excessive upregulation of CatX in the striatum and adjoining area on the ipsilateral side upregulation of CatX [68].

Osteoclasts are specialized cells that play a key role in the process of bone resorption through the breakdown of the bone mineral component and degradation of its organic matrix. CatK is the major collagenase in osteoclasts that mediates the inflammatory stress response on the bone surface [71]. The expression of CatK is regulated by RANKL (receptor activator of NF-κB ligand), and CatK promoter activator NFATc1 (nuclear factor of activated T cells). CatK is a very efficient collagenase. Collagen type I represents around 90% of bone material [72] so CatK was first proposed as a target for osteoporosis treatment [73]. Aside from this, since CatK also cleaves the main cartilage constituent type II collagen, as well as elastin, it is also implicated in pathologies such as RA, osteoarthritis, and cardiovascular diseases [74,75] (Figure 1). Numerous different CatK inhibitors have been generated and tested on different animal models, but only a few have reached clinical trials. Because of the species variations in CatK structure the initial compounds, which were efficient on rodent CatK, were largely inefficient on human CatK [76,77]. For that reason, cell-based models and the use of the ovariectomized cynomolgus monkey as a disease model [78] further helped in testing the CatK inhibitors, which have proved to have a major positive effect in decreasing bone resorption. Different CatK inhibitors have been evaluated in clinical trials for osteoporosis treatment and have shown a major reduction in bone resorption; however, none has passed into clinical practice. The most promising of these inhibitors was Relacatib, a highly selective non-basic compound developed by GlaxoSmithKline, which efficiently reduced serum and urine levels of degraded collagen products in a monkey model of osteoporosis but which was withdrawn from further trials because of its interference with the metabolism of other commonly used drugs. Odanacatib is a non-lysosomotropic nitrile that is highly selective for CatK (IC50 >300-fold lower compared to other cathepsins) [79]. The data from phase III clinical study showed that odanacatib administration led to an increase in bone density and reduced the risk of fractures [80]. Concomitantly, cardiovascular side effects, such as increased risk of stroke, were discovered and this led to the conclusion of further clinical testing [81]. The safety and tolerability phase of a clinical trial was also completed for another CatK inhibitor, MIV-7 11 [82], but there are no new data about its development (Figure 2). One of the most important issues with development of CatK inhibitors involves unwanted inhibition of other cathepsins (off-target effects) and the development to non-lysosomotropic and highly selective CatK inhibitors is a possible strategy to overcome this [82]. Another important problem is the effect of the inhibitors of CatK interfering with its other physiological functions (on-target effects) other than its function in bone degradation such as its involvement in kinin [83] and thyroid hormone [84] metabolism and TLR9 activation and signalling in antigen presenting cells [85]. Therefore, the most desirable CatK inhibitors will selectively block collagenolytic activity of CatK. Some of the possible strategies for creating such inhibitors are a blockade of secondary binding sites necessary for collagen binding [86,87], allosteric inhibitor development [88], and blocking the formation of CatK dimer, which is necessary for collagen degradation [72].
biomedicines-11-00476-t001_Table 1Table 1The role of cysteine cathepsins in inflammation and autoimmune diseases.CathepsinAutoimmune DiseasesInflammationCathepsin BApoptosis, inflammation, and infiltration of macrophages and TNF-α (+) cells to muscle tissue in polymyositis [66] Stimulates production and secretion of IL-1β in microglia [59,60]
CatB contributes to degradation of Aβ, thus helping maintain neuronal homeostasis [61,62], and increases accumulation of Aβ in Alzheimer’s disease [64,65]Cathepsin CExcessive neutrophil infiltration in GPA [45]Mediation of uncontrolled activity of NSPs [40,41]
Generation of proinflammatory chemokines [48] in activated microglia [47] that promote microglia M1 polarization and aggravate neuroinflammation [49]
Neuronal damage during inflammation or traumatic brain injury through stimulation of CCL2 and CXCL2 production from injured neurons and glia [51]Cathepsin KExcessive cleavage of main cartilage constituent type II collagen and elastin in RA, osteoarthritis and cardiovascular diseases [74,75], and
bone resorption [79]
Cathepsin L
May contribute to inflammatory responses in microglia [55]Cathepsin SMHC class II presentation of autoantigens [30]
Autoantibody production and renal deposition of immune complexes in SLE-related nephritis [31]
Increased T-cell responses towards autoantigents in Sjögren’s syndrome and increased cytokine secretion by CD14+ monocytes [35]
Infiltration of proinflammatory immune cells [39]
Cathepsin X
Secretion from microglia and astrocytes both in culture and in vivo in response to neuronal damage and inflammation [67,68,69,70]
Production of reactive oxygen species and proinflammatory cytokines [68]


### 2.2. Role of Cysteine Peptidases in Antitumor Immune Response and Their Targeting

In this section, we will review the role of cysteine cathepsins in the antitumor immune response (Table 2, Figure 3). Where applicable, we will also present the strategies of targeting cathepsins for the improvement of antitumor immune response currently in preclinical development. 

CatB was first proposed to have a protective role in preventing the self-destruction of cytotoxic lymphocytes upon secretion of cytotoxic granules [134], but this theory was later disputed [135]. Since then, the role of CatB in the antitumor immune response has been shown to be related to the processing of antigens [89]; reduced persistence of cytotoxic CD8+ T cells [90]; infiltration of immunosuppressive immune cells, such as tumor associated macrophages (TAM); myeloid-derived suppressor cells (MDSCs) and regulatory T cells (Tregs) in gliomas [91] and pancreatic ductal adenocarcinoma [92]. Additionally, commonly used chemotherapeutics such as 5-fluoro uracil and gemcitabine, have been shown to cause the release of CatB from lysosomes, which activated the inflammasomes. This activated caspase-1 and caused release of IL-1β and IL-17 from MDSCs, which limited the efficacy of anticancer treatment [99,100]. Recently, exciting research has shown a new therapeutic strategy of exploiting high CatB expression in tumor tissue. Nanoparticles were developed in the form of a prodrug containing a CatB cleavable peptide fragment. Nanoparticles accumulate in the tumor tissue, where CatB cleavage releases doxorubicin and photosensitizer, causing immunogenic cell death upon visible light irradiation. This converted immunologically cold TME, as DAMP signals promoted dendritic cell (DC) maturation and T cell activation. This treatment strategy could be used in synergy with checkpoint-blockage immunotherapy [93,94]. Indeed, addition of the anti-PD-L1 peptide to the prodrug nanoparticle efficiently inhibited tumor progression with minimal side effects in the breast-tumor mouse model [95]. A similar strategy to reprogram the immunosuppressive tumor microenvironment (TME) was used by other groups. A CatB-cleavable segment was used to deliver a COX1/2-containing nanoparticle to the tumor microenvironment. Persistent degradation of COX1/2 depleted the metabolite PGE2, which is responsible for the activation of immune suppressor cells [96]. In another strategy, CatB in TME was used to release miRNA which downregulated PD-L1 expression from the nanoparticle [97]. Other prodrug antibodies targeting cathepsins can be found reviewed in [2,98].

CatC, together with CatH, is responsible for post-translational processing of granule serine peptidases found in cytotoxic-immune cells [101,102]. CatC and CatH are essential for the activation of granzymes A and B which are the mediators of the granule-dependent cytotoxic pathway in NK cells and CTLs [136]. Both cell types are important for the elimination of cancer cells. CTLs mediate their effects towards differentiated tumor cells, after successful antigen presentation. On the other hand, NK cells eliminate undifferentiated tumor stem cells without the need for antigen priming [137,138]. Additionally, NK cells stimulate the adaptive immune response by releasing cytokines such as IFN-γ [139,140]. CatC was also found to be involved in the promotion of the metastasis of breast cancer to the lungs. Tumor-derived CatC facilitated IL-1β processing and NFκB activation, which upregulated IL-6 and CCL3 and caused neutrophil recruitment. Neutrophils supported the metastatic growth of cancer cells by production of reactive oxygen species and formation of NETs [103].

Bioinformatics showed that increased CatF expression in lung cancer contributed to favorable patient prognosis. CatF was shown to be expressed predominantly in macrophages and its expression was positively related to infiltration of other immune cells such as B cells, DCs, CD8, and CD4+ T cells and NK cells. The study speculated that CatF might function as a tumor-suppressor gene via contributing to antigen presentation for the antitumor immune response [104]. Validation of the study using in vitro and in vivo methods is needed. 

In addition to CatC, CatH has also been confirmed to be involved in the activation of granzymes in cytotoxic immune cells. In colorectal cancer patients its protein levels were found elevated in serum, [141] and moreover, CatH gene was identified as one of five colorectal-cancer-specific immune genes. The bioinformatics study speculated that CatH is involved in MHC class II antigen presentation [105].

Knowledge about CatK involvement in antitumor immune response is limited. It was shown that CatK derived from bone marrow macrophages is critical for the progression of prostate cancer in the bone via CCL2 and COX2 pathways [106]. CatK was also found to promote metastasis of squamous cell carcinoma. It was identified as the downstream target of the CD200- CD200R axis, with MDSC-like cells and TAMs being the main source of CatK [142]. Another study which implicated CatK in cancer progression showed that imbalance of gut microbiota upregulates CatK expression in colorectal cancer. After binding to TLR4, CatK stimulated polarization of M2 TAMs, which started secreting IL-10 and IL-17, and therefore contributed to the invasion and metastasis of colorectal cancer [107]. 

Another cathepsin implicated in the regulation of the granule-mediated cytotoxicity is CatL. It was shown that CatL processes perforin-1; however, it is not the only enzyme capable of perforin activation [108]. CatL was also shown to contribute to the resistance of melanoma cells to complement-mediated lysis [109] and the ability of DC to present antigens to T cells in colorectal cancer patients [110]. Additionally, when breast-cancer cells were used to generate MDSCs from healthy human monocytes, the levels of CatL and CatX increased, indicating that tumor–immune-cell interactions are important for the evaluation of anti-cancer potential of anti-cathepsin treatments [111].

CatS has an important role in anticancer immunity. It can be beneficial as it is important for the MHC II-mediated antigen presentation, but it is also involved in the polarization of APCs from M1 to M2 phenotype which supports MDSCs and TAMs, and enhanced proliferation of Tregs as well as tumor proliferation and invasion [112,113,114,115,116,117]. Therapeutic antibodies were developed in order to target CatS and block its invasion and angiogenic properties in colorectal-carcinoma cell lines and xenograft tumor models [143]. Additionally, as CatS is also expressed on the surface of colorectal-tumor cells the binding of the anti-CatS antibody Fsn0503 facilitated antibody-dependent cell-mediated cytotoxicity [144]. In another study, a small molecular inhibitor of CatS decreased expansion of Tregs and enhanced CD8+ cytotoxic T-cell numbers in a mouse model of bladder carcinoma [145] and non-Hodgkin lymphoma [146]. Several other small-molecular inhibitors have been developed in the last decade and tested in preclinical settings for their selectivity and anticancer properties and are reviewed in [147]. As CatS is secreted in the TME by both tumor and immune cells it is also a promising target for drug delivery. A liposomal drug carrier functionalized with stefin A was shown to be able to selectively target cathepsins both in vitro and in vivo [118].

The role of CatV in cancer is not yet well understood. Increasing number of studies show elevated expression levels of CatV in several cancers, where it might be involved in cancer-progression processes (recently reviewed in [119]). CatV might also be involved in attenuation of antitumor immune response. It was shown to activate CysF, an endogenous inhibitor of CatC, H and L which are involved in the activation of granule mediated cytotoxicity of cytotoxic immune cells [120]. Recently, new inhibitors of CatV were designed and were shown to impair tumor-cell proliferation and elastin degradation and to be beneficial in the antitumor immune response by preventing the activation of CysF [148].

Recently, the CatW gene-expression profile was shown to correlate with increased overall survival in endometrial-cancer patients. Furthermore, CatW expression positively correlated with the infiltration of B cells, DC, macrophages, and CD4+ T cells to the tumor site [121]. The function of CatW in the immune response to cancer remains unknown. Even though CatW was shown to be predominantly expressed by cytotoxic immune cells [149,150], and even upregulated after stimulation of NK cells with IL-2 [151], its function seems unrelated to the cytotoxic function. Cytotoxicity of immune cells from CatW-deficient mice remained unchanged [152]. Some studies have shown that its levels decrease during cytotoxic attack of NK-92 cells towards the K-562 cell line [153,154], and that CatW is secreted during target-cell killing [155]. The location of CatW location also differs from that of other cathepsins as it was found in the endoplasmic reticulum from where it can be secreted via Golgi apparatus and secretory vesicles [151,155]. 

CatX was shown to promote tumor invasion [127,128], epithelial-to-mesenchymal transition [130], and cleavage of tumor suppressor profilin-1 [132], and to contribute to resistance to apoptosis in several tumor types [133]. It also has several important functions in the cells of the immune system. Due to its interaction with β2 integrin receptors it is important for the migration of T cells and the formation of the immunological synapse in T cells [122,123,124,125] but not NK cells [126], phagocytosis of macrophages [129], and adhesion-dependent maturation of DCs [131]. Even though new inhibitors of CatX show promising results in reducing tumor progression both in vitro and in vivo [156], it is important to also evaluate the effects of CatX inhibitors in the context of antitumor immune responses.

## 3. Challenges and the Future of Cathepsins as Therapeutic Targets

Although development of new technologies, such as quantitative proteomics and in-vivo imaging, as well as an extensive use of in-vivo models have enabled major expansion of protease research in the past decade, surprisingly few protease inhibitors have reached the phase of clinical studies in their development and testing and even fewer have been approved and entered clinical practice. Their ubiquitous presence and involvement in crucial physiological processes, as well as their overlapping substrate specificities, are largely the cause of poor translation to clinics. Furthermore, even in cases where significant selectivity and almost non-existent off-target effects were achieved in animal models, drastically different effects were recorded when these inhibitors were tested on human subjects due to species-specific differences in cysteine-cathepsin structure and specificity. Although development of highly selective cathepsin inhibitors has been achieved with recent technological advancements, the major challenge remains the complexity of immunological processes that cysteine cathepsins are involved in. The most promising strategy for conquering this challenge is targeted drug delivery; an example of is the use of antibody–drug conjugates (ADCs) to cancer cells. By solely targeting extracellular cathepsins in the disease microenvironment for successful drug delivery, unwanted on-target effects can be avoided. Another approach is targeting membrane antigens/receptors expressed solely on certain cell types (such as cancer cells). Upon binding to such receptors, ADCs are internalized and release their conjugates into the endo/lysosomes which is rich with cysteine cathepsins. Finally, considering that multiple cysteine cathepsin inhibitors showed promising results in clinical trials, but were discontinued due to unexpected side-effects and interactions with other drugs, data gained from clinical trials represents another promising alternative for future development of cysteine-cathepsin inhibitors. 

## Figures and Tables

**Figure 1 biomedicines-11-00476-f001:**
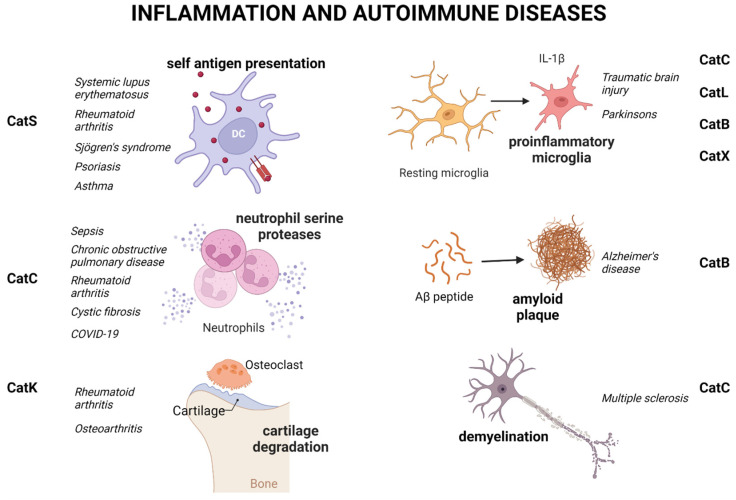
Cysteine cathepsin involvement in inflammation and autoimmune diseases. Cysteine cathepsins are involved in the presentation of self-antigens via MHC II molecules on antigen presenting cells, dysregulation of neutrophil serine proteases, and cartilage degradation. These processes detrimental for several autoimmune diseases. On the other hand, cysteine cathepsins also participate in the generation of inflammation in the brain and therefore impact the progression of several neurodegenerative diseases. Created with Biorender.

**Figure 2 biomedicines-11-00476-f002:**
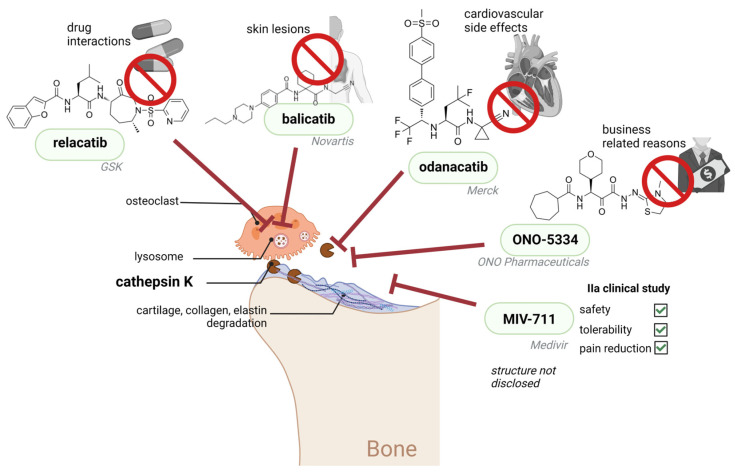
Several cathepsin K inhibitors have entered clinical trials. Lysosomotropic cathepsin K inhibitors relacatib, and balicatib were discontinued due to the unexpected effect on metabolism of commonly used drugs, and side-effect occurrence of morphea-like skin lesions, respectively. Nonlysosomotropic inhibitor odanacatib was also discontinued due to cardiovascular side-effects. ONO-5334 compound had a better safety profile; however, its development was stopped due to business-related reasons. Another cathepsin K inhibitor, MIV-711, with currently undisclosed structure and mechanism of action, was found safe and tolerable and showed clinically beneficial effects in a phase IIa clinical study; however, there is currently no further information about the progress on the clinical testing for MIV-711. Red T-bar arrow represents modes of cathepsin K inhibition of presented compounds, either in the lysosomes or in the extracellular space. Created with Biorender.

**Figure 3 biomedicines-11-00476-f003:**
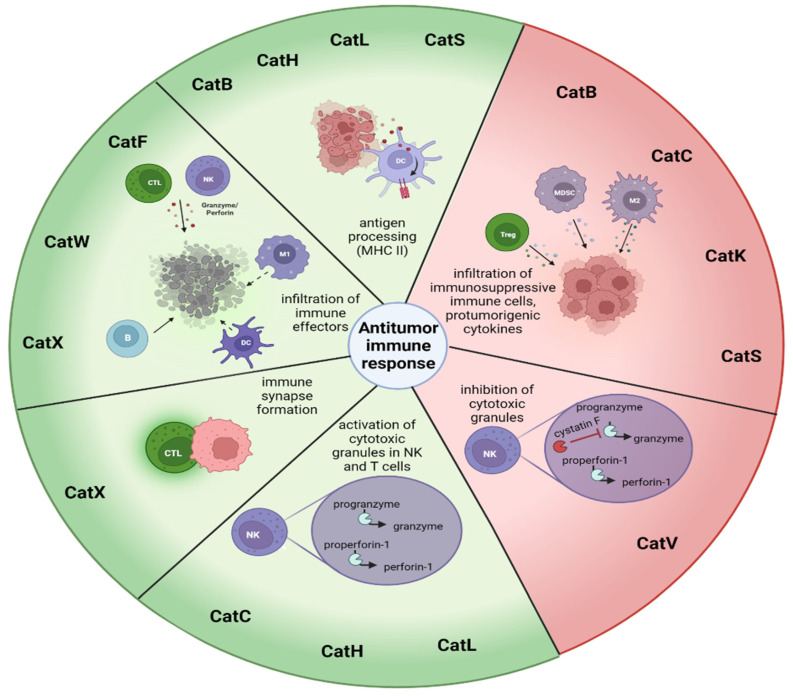
Cysteine cathepsin involvement in antitumor immune response. Cysteine cathepsins are involved in several beneficial as well as negative processes in antitumor immunity. On the one hand, they facilitate presentation of antigens, which encourages the infiltration of effector immune cells to tumors. Additionally, they participate in the immune-synapse formation between tumor and cytotoxic T cells and activate the effector molecules in the cytotoxic granules of NK cells and CTLs. On the other hand, they support the infiltration of immunosuppressive immune subsets such as MDSCs, Tregs and TAMs, and can prevent the activation of cytotoxic granules of NK cells and CTLs. Red T-bar arrow indicates inhibitory pathway, while black arrows indicate activation pathways (curved) or movements of the cells (straight). Created with Biorender.

**Table 2 biomedicines-11-00476-t002:** The role of cysteine cathepsins in immune response to cancer.

	Immune Response to Cancer	
Cathepsin	Stimulating Antitumor Immune Response	Decreasing Antitumor Immune Response
Cathepsin B	Antigen processing [89]	Reduced persistence of CD8+ T cells [90], infiltration of immunosuppressive TAMs, MDSCs, and Tregs [91,92]
	Target for (pro)drug delivery [2,93,94,95,96,97,98]	Activation of the inflammasome under the influence of chemotherapeutics, and IL-1β and IL-17 secretion from MDSCs [99,100]
Cathepsin C	Activation of granule serine peptidases in cytotoxic immune cells [101,102]	Promotion of metastasis, by neutrophil recruitment, production of reactive oxygen species, formation of NETs and secretion of IL-1β, IL-6, and CCL3 [103]
Cathepsin F	Infiltration of B cells, DCs, CD8 and CD4+ T cells, and NK cells [104]	
Cathepsin H	Activation of granule serine peptidases in cytotoxic immune cells [101,102]	
	MHC class II antigen presentation [105]	
Cathepsin K		Bone metastasis [106]
		Polarization of M2 TAMs, secreting IL-10 and IL-17 [107]
Cathepsin L	Activation of perforin-1 [108]	Resistance to complement-mediated lysis [109]
	Antigen presentation [110]	Increased in MDSCs [111]
Cathepsin S	MHC II-mediated antigen presentation [112]	Polarization of APCs to M2 phenotype, supporting enhanced proliferation of MDSCs and TAMs, Tregs [112,113,114,115,116,117]
	Target for drug delivery [118]	
Cathepsin V		Cancer progression [119]
		Activation of CysF and decreasing cytotoxicity of NK cells and CD8+ T cells [120]
Cathepsin W	Infiltration of B cells, DC, macrophages, and CD4+ T cells to the tumor site [121]	
Cathepsin X	Migration of T cells and formation of immunological synapse in T cells [122,123,124,125] but not NK cells [126]	Tumor invasion [127,128]
	Phagocytosis of macrophages [129]	Epithelial-to-mesenchymal transition [130]
	Adhesion-dependent maturation of DCs [131]	Cleavage of tumor suppressor profilin-1 [132]
		Resistance of apoptosis in several tumor types [133]
		Increased in MDSCs [111]

## Data Availability

Not applicable.

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
