# Peer review of "Cysteine Cathepsins as Therapeutic Targets in Immune Regulation and Immune Disorders"

_biomedicines, 2023, doi:10.3390/biomedicines11020476_

Round 1

Reviewer 1 Report

In this review paper entitled “Cysteine cathepsins as therapeutic targets in immune regulation and immune disorders” aims to perform an overview of cysteine cathepsin involvement and possible targeting options for mitigation of aberrant function in immune disorders such as inflammation, autoimmune diseases, and immune response in cancer. Although well written, it is not clear the added value compared to others DOI: 10.1080/14728222.2020.1746765, or DOI: 10.1080/07391102.2022.2135603, or DOI: 10.1016/j.mam.2022.101106. a table resume integrating the most relevant features of biological process per each one could be presented. A perspective of function associated to structure could also illustrated. Additionally, research based on clinical trials should be pointed as alternative for future studies. 

Author Response

We thank the reviewer for the opportunity to improve our article. In this paper we present - the current knowledge on cysteine cathepsin inhibitors, summarizing and adding also the most UpToDate data on the current development in the field, focusing on inflammatory and autoimmune diseases as well as the role and the potential of cysteine cathepsin inhibitors in improving antitumor immune response. Added value of our review is summarizing the state of the development of the inhibitors for all of cysteine cathepsins.

As reviewer suggested we have added additional tables, summarizing the role of each cysteine cathepsin in Inflammation and Autoimmune diseases (Table 1) and Immune response to cancer (Table 2).

We agree with the reviewer that a good base for future studies should be research, based on clinical trials and we have added this statement to the conclusion section.

Reviewer 2 Report

The review titled:Cysteine cathepsins as therapeutic targets in immune regulation and immune disorders handled different points about the usage of cysteine cathepsins as a possible legends for treatment inflammatory and immune disorderes.

From my points of view this study collected many works that show the down expression of cysteine cathepsins sub-types may be of promising effect for amelioration of inflammation and neuro-degeneration. The most trials discussed by authors are discontinued at Phase I, II or III trials  Why.

You need to add more figures  to outline your points in discussed sections.

Author Response

We thank the reviewer for the opportunity to improve our article.

Indeed, a very small number of cysteines cathepsin inhibitors entered clinical trials and were approved for clinical treatment. For some of them, such as cathepsin S inhibitors, clinical trials were terminated in Phase II, the efficacy phase, due to requirement for high-level inhibition of enzyme activity needed to achieve therapeutically significant effects, most likely as a consequence of the Cat S species-specific differences between mice and humans (as discussed in the text, section 2. Paragraph 2, line 107-110). In addition, for cathepsin C inhibitors compensatory role of other cathepsins such as Cathepsin H and off-target effect such as major lack of processing of multiple granule serine peptidases, led to termination of numerous clinical studies (as discussed in the text, section 2. Paragraph 3, line 142-145). For cathepsin K, which is one of the most studied cathepsins, several highly selective inhibitors entered clinical trials, but were terminated due to cardiovascular side effects, such as increased risk of stroke, due to the unexpected effect on metabolism of commonly used drugs, or occurrence of morphea-like skin lesions (as discussed in the text, section 2. Paragraph 9, line 112-124).

An additional figure (now Figure 2) was added to further outline this point.

Round 2

Reviewer 1 Report

It is acceptable for publication.